

# The role of TRIF protein in regulating the proliferation and antigen presentation ability of myeloid dendritic cells through the ERK1/2 signaling pathway in chronic low-grade inflammation of intestinal mucosa mediated by flagellin-TLR5 complex signal

Zhaomeng Zhuang[1,2], Yi Chen[1], Juanhong Zheng[1] and Shuo Chen[1]

[1] Gastroenterology, Zhejiang Chinese Medical University Affiffiliated Wenzhou Hospital of Integrated Traditional Chinese and Western Medicine, Wenzhou, Zhejiang, China
[2] The First Clinical Medical College, Zhejiang Chinese Medical University, Hangzhou, Zhejiang, China

Corresponding author
Shuo Chen, zzmandmm@126.com

## ABSTRACT

**Objective**. The objective is to explore whether the flagellin-TLR5 complex signal can enhance the antigen presentation ability of myeloid DCs through the TRIF-ERK1/2 pathway, and the correlation between this pathway and intestinal mucosal inflammation response.

**Methods**. Mouse bone marrow-derived DC line DC2.4 was divided into four groups: control group (BC) was DC2.4 cells cultured normally; flagellin single signal stimulation group (DC2.4+CBLB502) was DC2.4 cells stimulated with flagellin derivative CBLB502 during culture; TLR5-flagellin complex signal stimulation group (ov-TLR5-DC2.4+CBLB502) was flagellin derivative CBLB502 stimulated ov-TLR5-DC2.4 cells with TLR5 gene overexpression; TRIF signal interference group (ov-TLR5-DC2.4+CBLB502+Pepinh-TRIFTFA) was ov-TLR5-DC2.4 cells with TLR5 gene overexpression stimulated with flagellin derivative CBLB502 and intervened with TRIF-specific inhibitor Pepinh-TRIFTFA. WB was used to detect the expression of TRIF and p-ERK1/2 proteins in each group of cells; CCK8 was used to detect cell proliferation in each group; flow cytometry was used to detect the expression of surface molecules MHCI, MHCII, CD80, 86 in each group of cells; ELISA was used to detect the levels of IL-12 and IL-4 cytokines in each group.

**Results**. Compared with the BC group, DC2.4+CBLB502 group, and ov-TLR5-DC2.4+CBLB502+Pepinh-TRIFTFA group, the expression of TRIF protein and p-ERK1/2 protein in ov-TLR5-DC2.4+CBLB502 group was significantly upregulated (TRIF: $p = 0.02, = 0.007, = 0.048$) (ERK1: $p < 0.001, = 0.0003, = 0.0004; ERK2: p = 0.0003, = 0.0012, = 0.0022$). The cell proliferation activity in ov-TLR5-DC2.4+CBLB502 group was enhanced compared with the other groups ($p = 0.0001, p < 0.0001, p = 0.0015$); at the same time, the expression of surface molecules MHCI, MHCII, CD80, 86 on DCs was upregulated ($p < 0.05$); and the secretion of IL-12 and IL-4 cytokines was increased, with significant differences (IL-12: $p < 0.0001,$

**PeerJ** ___________________________________________________________

$p < 0.0001$, $p = 0.0005$; IL-4: $p = < 0.0001$, $p = < 0.0001$, $p = 0.0001$). However, the ov-TLR5-DC2.4+CBLB502+Pepinh-TRIFTFA group, which was treated with TRIF signal interference, showed a decrease in intracellular TRIF protein and p-ERK1/2 protein, as well as a decrease in cell proliferation ability and surface stimulation molecules, and a decrease in the secretion of IL-12 and IL-4 cytokines ($p < 0.05$).

**Conclusion**. After stimulation of flagellin protein-TLR5 complex signal, TRIF protein and p-ERK1/2 protein expression in myeloid dendritic cells were significantly up-regulated, accompanied by increased proliferation activity and maturity of DCs, enhanced antigen presentation function, increased secretion of pro-inflammatory cytokines IL-12 and IL-4. This process can be inhibited by the specific inhibitor of TRIF signal, suggesting that the TLR5-TRIF-ERK1/2 pathway may play an important role in abnormal immune response and mucosal chronic inflammation infiltration mediated by flagellin protein in DCs, which can provide a basis for our subsequent animal experiments.

# INTRODUCTION

The TRIF protein (Toll/IL-1R-domain-containingadaptor-inducing IFN- $\beta$) is an important pathway protein for intracellular transmission of Toll-like receptors (TLRs) signaling (*Yamamoto et al., 2003*), and studies have shown that overexpression of TRIF protein in the intestinal mucosa can activate downstream proliferation regulatory protein ERK1/2 signaling. It plays an important role in intestinal mucosal immune regulation and chronic inflammatory infiltration (*Guindi et al., 2018*; *Jiang et al., 2018*).

Our previous research results have shown that in some diseases related to intestinal dysbiosis, there is an increase in intestinal flagellate colonization and mucosal flagellin protein content, which induces high expression of TLR5 protein and TRIF protein in colon mucosa. This phenomenon can promote the proliferation and antigen presentation function of lamina propria dendritic cells (LPDCs) in the intestinal mucosal inner layer, as well as mucosal inflammatory factor infiltration (*Zhuang et al., 2022*; *Li et al., 2015*; *Zhuang et al., 2016*).

Flagellin protein is the main component of the flagellar structure of flagellated bacteria. Intestinal flagellated bacteria play an important role in the microbial colonization, secondary infection, and regulation of intestinal mucosal immune response due to their unique flagellar structure. Pathogenic flagellated bacteria mainly belong to the $\gamma$ branch of Proteobacteria, including Escherichia, Klebsiella, Pseudomonadales, Salmonella, *etc.* The highly conserved TLR recognition site on the flagellar structure can specifically bind to Toll-like receptor 5 (TLR5) on the dendritic cells of the lamina propria, thereby playing an important role in the immune response against pathogenic bacteria in the intestinal mucosa and leading to chronic inflammatory reactions in the intestinal mucosa (*Mohari et al., 2018*).

Toll-like receptor 5 (TLR5) is a specific receptor for bacterial flagella and plays an extremely important role in intestinal anti-infection immunity. However, it is currently unclear whether TRIF protein plays a role in the abnormal immune function of dendritic cells mediated by flagellin-TLR5 signaling. Whether this process can activate the downstream proliferation activating protein ERK1/2 has not yet attracted the attention of researchers.

Based on our previous research results, we speculate that TRIF protein may play an important role in the proliferation, antigen presentation ability maturation, and secretion of inflammatory factors in dendritic cells during the occurrence of abnormal immune responses mediated by flagellin protein in colon mucosa. In order to further verify that the flagella protein of flagella can promote dendritic cell activation and functional maturation through the TLR5 receptor on dendritic cells, and this process can be activated through the TRIF adapter protein in dendritic cells, we designed this cell experiment.

## MATERIALS AND METHODS

### Experimental cell line and lentivirus

Mouse bone marrow-derived dendritic cell line DC2.4 (purchased from OTWO). Culture conditions: cells were cultured in EMEM medium containing 10% premium fetal bovine serum and 1% penicillin-streptomycin double antibiotics The medium was changed every 2–3 days, and the cells were passaged at a ratio of 1:3 when they reached confluency. Cells were cultured at 37 °C, 5% CO2, and saturated humidity. Experimental lentivirus (purchased from Wisent Biotechnology, Quebec, Canada).

### Experimental instruments and reagents

#### Main instruments

AB Step One plus Real Time PCR System (Applied Biosystems AB, Waltham, MA, USA); Stabilized DNA electrophoresis apparatus (Bio-Rad, Hercules, CA, USA); Trace DNA/RNA quantification instrument (Thermo Fisher Scientific, Waltham, MA, USA); Tanon-1600 gel imaging system (Tanon, Shanghai, China); Amicon Ultra-15 100 KDa concentrator (Millipore, USA); Flow cytometer (Beckman, USA); Multifunctional ELISA reader Varioskan LUX (Thermo Fisher Scientific).

#### Main reagents

RNAiso plus 9109 Q (TaKaRa Tokyo, Japan); Hifair® III 1st Strand cDNA Synthesis Kit (gDNA digester plus)11139 ES 10 (YEASEN, Shanghai, China); Hieff UNICON® Universal Blue Qpcr SYBR Green Master Mix 11184 ES 03 (YEASEN); Recombinant flagellin protein derivative CBLB502 (InvivoGen, Toulouse, France); TIRF-specific antagonist Pepinh-TRIFTFA (MCE, Sichuan, China); EMEM culture medium (ATCC, Manassas, VA, USA); DMSO (Sigma-Aldrich, St. Louis, MO, USA); Fetal bovine serum (Gibco, Waltham, MA, USA); Plasmid mini extraction kit (Omega, Norcross, GA, USA); Plasmid maxi extraction kit (Omega, Norcross, GA, USA); EcoRI (TaKaRa, Japan); BamHI (TaKaRa); RNAisoplus (TaKaRa); TLR5 antibody (rabbit source; Proteintech, Rosemont, IL, USA); TRIF antibody (rabbit source, Affinity, USA); P-ERK1/2 antibody (rabbit source; Abcam,

Waltham, MA, USA); CD80 (B7-1) Monoclonal Antibody, PE (eBioscience, San Diego, CA, USA); CD86 (B7-2) Monoclonal Antibody, PE (eBioscience, San Diego, CA, USA); MHC Class I (H-2Kb) monoclonal antibody, PE (eBioscience, San Diego, CA, USA); MHC Class II (I-A/I-E) monoclonal antibody, PE (eBioscience); Cell Counting Kit (Beyotime Biotechnology Co., Ltd., Haimen, China); rat interleukin 4 (IL-4) ELISA (Jiangsu Jingmei Bioengineering Co., Ltd., Guangzhou City, China); rat interleukin 12 (IL-12) ELISA kit (Jiangsu Jingmei Bioengineering Co., Ltd., Guangzhou City, China).

## Construction of overexpression lentiviral vector and infection of target cells

Specific sticky end sequences were obtained by splicing by overlap extension(SOE) PCR. The target vector pLVX-mCMV-zsGreen-puro was digested with enzymes. The purified synthetic product was connected with linearized vector, and the connected product was transformed into bacterial competent cells. The monoclonal clones were sequenced and identified, and the sequencing results were compared and analyzed. The sequence that was completely correct was the successfully constructed target gene lentiviral expression plasmid vector, named TLR5-pLVX-zsGreen-puro. Sufficient overexpression vector plasmid was obtained by ultra-pure endotoxin extraction of the constructed lentiviral overexpression vector plasmid. Lentiviral transfection was used to transfect cells, screen stable transfected cells continuously and pass three generations, and qRT-PCR and Western blot was used to verify the TLR5 gene transfection efficiency for subsequent experiments.

## Experimental methods
### Grouping
Divide mouse bone marrow-derived dendritic cell line DC2.4 into four groups.

Normal control group (BC): DC2.4 cells in normal culture group;

Single signal group of flagellin protein (DC2.4+CBLB502): DC2.4 cells were stimulated with 100ng/mL of flagellin protein derivative CBLB502.

TLR5-flagellin protein complex signal stimulation group (ov-TLR5-DC2.4+CBLB502): After TLR5 gene overexpression treatment of DC2.4 cell line, ov-TLR5-DC2.4 cells were established and stimulated with 100ng/mL of flagellin protein derivative CBLB502.

TRIF signal inhibition group (ov-TLR5-DC2.4+CBLB502+Pepinh-TRIFTFA): After stimulating ov-TLR5-DC2.4 cells with 100ng/mL of flagellin protein derivative CBLB502, TRIF-specific inhibitor Pepinh-TRIFTFA at 40 μM was added for intervention. See Fig. 1 for details.

### Cell culture
Mouse bone marrow-derived dendritic cell line DC2.4 was cultured in EMEM medium containing 10% FBS and 1% penicillin-streptomycin double antibody. It was cultured in a cell culture incubator at 37 °C, 5% $CO_2$, and 100% relative humidity; the cells adhered to the wall and were passaged for subsequent experiments.
DC2.4

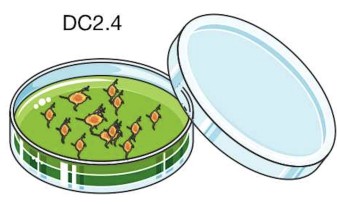

Black control group

DC2.4+100 ng/mL CBLB502

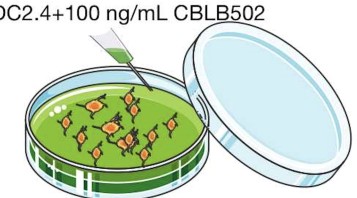

Flagellin single signaling group

ov-TLR5-DC2.4+100 ng/mL CBLB502

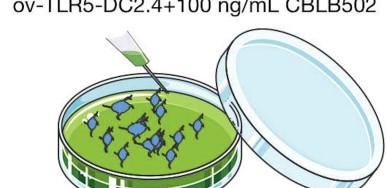

TLR5-Flagellin single stimulation group

ov-TLR5-DC2.4+100 ng/mL CBLB502
+ 40 µM Pepinh-TRIF-TFA

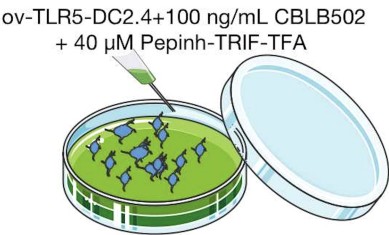

TRIF single suppression group

**Figure 1  Cell grouping and drug intervention.**

## QT-PCR

Validation of ov-TLR5-DC2.4 cell line TLR5 gene transfection efficiency: internal reference and target gene primers were designed according to the gene accession number published by NCBI:

$\beta$-actin-F GATATCGCTGCGCTGGTCG 132 bp
$\beta$-actin-R CATTCCCACCATCACACCCT 132 bp
TLR5-F CCACATCATGGGGGTTCTGGCT 100 bp
TLR5-R AAGGTCCAGTTGTAGCACCG 100 bp

A total of 1 ml of RNAiso was added to the cell samples. A homogenate was made and the supernatant was centrifuged. Isopropanol was added to precipitate the RNA. The supernatant, and 1 ml of 75% ethanol was added, centrifuged for 5 min, dried at room temperature. The appropriate amount of RNase-free water was added to dissolve the precipitate, and then an ultra-micro-spectrophotometer was used to determine the quantity and concentration of the RNA. The quality and concentration of RNA were determined by ultra-micro spectrophotometer. qPCR was performed after reverse transcription of RNA, and the reverse transcription reaction system was prepared as follows: Hieff UNICON Univeral Blue Qpcr SYBR Green Master Mix 5 µL; forward and reverse primers (10 µM each) 0.4/0.4 µL; Template DNA/cDNA 1 µL; Template DNA/cDNA 1 µL; Template DNA/cDNA 1 µL; Template DNA/cDNA 1 µL. cDNA 1 µL; RNase Free ddH2O Up to 10 µL. Reverse transcription reaction conditions were set: 95 °C for 2 min for 1 cycle, 95 °C for 10 s for 40 cycles, 60 °C for 30 s for 40 cycles. The final plasmid copy number was calculated: slope−3.219; Y-Inter: 7.349; $R^2$: 0.994; Eff%: 104.482.

### Drug intervention

Recombinant flagellin protein derivative CBLB502 for mice: 10 ug powder dissolved in 0.1 ml DMSO. A stock solution was prepared with a concentration of 100 ug/mL for later use; TRIF-specific inhibitor Pepinh-TRIFTFA: A total of 500 ug of powder was dissolved in 0.0374 mL of DMSO to prepare a stock solution with a concentration of 50 mM.

(1) Cells were resuspended and plated in separate dishes for each group until the cell fusion degree reaches about 80%; (2) Different doses of drugs were added according to the grouping information mentioned above; (3) and incubated in a culture box for 24 h.

### WB detection of TRIF and p-ERK1/2 protein expression in each group of cells

Add 0.1 ml cell lysis buffer to lyse cells in every 106 cells. Then, after preparing the BSA standard curve, each group sample was tested. A total of 100 $\mu$L of the diluted test sample was added to the microplate, with three parallel samples for each sample, and then 100 $\mu$L of working solution was added and mixed. The reaction was carried out at 37 °C for 60 min and then cooled to room temperature. Finally, the spectrophotometer was set to a wavelength of 562 nm for measurement. The protein concentration of the test sample was calculated based on the standard curve by subtracting the average absorbance value of the blank from that of each sample solution, and SDS-PAGE electrophoresis and exposure were performed.

### CCK8 assay was used to detect the proliferation of dendritic cells in each group

After drug intervention, the cells were tested at three time points of d3, d5, and d7, and four sub-wells were set up for each group of cells. A total of 50 $\mu$L of CCK-8 solution was added to each well, and the culture plate was incubated in a culture box for 3 h. After incubation, the plate was evenly shaken and 100 $\mu$L was taken from each well and placed in a 96-well plate. The absorbance value at 450 nm was measured using an enzyme-linked immunosorbent assay (ELISA) reader. Cell absorbance value = sample well absorbance value - blank control well absorbance value.

### Flow cytometry was used to detect the expression of surface MHCI molecules, MHCII molecules, CD80, and CD86 in each group of DCs

The cells in each group were treated with trypsin, centrifuged to obtain cell pellets, and resuspended in approximately 1 mL of pre-cooled PBS. After centrifugation, the cells were resuspended in 400 ul of PBS (approximately 2 × 106 cells) and divided into four sample tubes, as well as a control tube. CD80/PE (1:200), CD86/PE (1:100), MHCClassI/PE (1:80), and MHCClassII/PE (1:200) flow cytometry antibodies were added to each of the four sample tubes and incubated at room temperature in the dark for 20 min. The tubes were then centrifuged at 1,000 rpm at 4 °C for 5 min, and the supernatant was discarded. This washing process was repeated three times. Finally, cells were resuspended in 100 $\mu$L PBS and analyzed by flow cytometry.

### IL-12 and IL-4 cytokine levels were detected by ELISA in each group

Purified rat IL-12 and IL-4 capture antibodies were coated onto microplates to make solid-phase antibodies using IL-12 and IL-4 ELISA kits. IL-12 and IL-4 were added to the

coated microplates, followed by binding with HRP-labeled detection antibodies to form an antibody-antigen-enzyme-labeled antibody complex. After thorough washing, TMB substrate was added for color development. The absorbance (OD value) was measured at 450 nm wavelength using an ELISA reader, and the IL-12 and IL-4 content were calculated in the sample based on the standard curve.

### Statistical methods

Quantitative data are expressed as mean ± standard deviation, and qualitative data are expressed as frequency (n) and percentage (%). Depending on the specific situation after testing for data independence, normality, and homogeneity of variance, single-factor analysis of variance (ANOVA) or Kruskal–Wallis test is used for within-group comparison. When there is within-group difference, Bonferroni or Tukey *post-hoc* test is used for pairwise comparison between groups. Data statistical analysis is performed using Graphpad Prism software (GraphPad Software, La Jolla, CA, USA), and figures are made using Graphpad Prism and Figdraw software. $P < 0.05$ is considered statistically significant.

## RESULTS

### Construction of ov-TLR5-DC2.4 cell line

After infecting DC2.4 cells with lentivirus for 48–96 h, the efficiency of infecting target cells was >80% as observed under a fluorescence microscope. Samples were collected for Western blot and qRT-PCR analysis of TLR5 protein and RNA expression in each group.

The results showed that the expression of TLR5 protein and gene in ov-TLR5-DC2.4 cell line was up-regulated compared with DC2.4 cell line and empty virus-transfected DC2.4 cell line, and the difference was statistically significant ($p < 0.0001$). These results indicate that the construction of ov-TLR5-DC2.4 cell line was successful and can be used for subsequent experiments. See Figs. 2B–2C for details.

### Expression of TRIF and p-ERK1/2 proteins in each group of dendritic cells

Western-blot was used to detect the expression of TRIF protein and phosphorylated ERK1/2 protein in dendritic cells of each group.

The results showed that the intracellular expression of TRIF protein was up-regulated in ov-TLR5-DC2.4+CBLB502 group compared with blank control group, DC2.4+CBLB502 group and ov-TLR5-DC2.4+CBLB502+Pepinh-TRIFTFA group ($p = 0.02$; $p = 0.007$; $p = 0.048$); see Fig. 3A for details.

Meanwhile, the downstream functional protein p-ERK1/2 of TRIF was significantly up-regulated in ov-TLR5-DC2.4+CBLB502 group, which was statistically significant compared with the other three groups (ERK1: $p < 0.001$; $p = 0.0003$; $p = 0.0004$; ERK2: $p = 0.0003$; $p = 0.0012$; $p = 0.0022$); see Fig. 3B for details.

These results suggest that flagellin-TLR5 combined signal stimulation can promote the high expression of TRIF protein in DC2.4 cells, and at the same time increase the expression of downstream functional protein p-ERK1/2 protein. When the intracellular TRIF signal

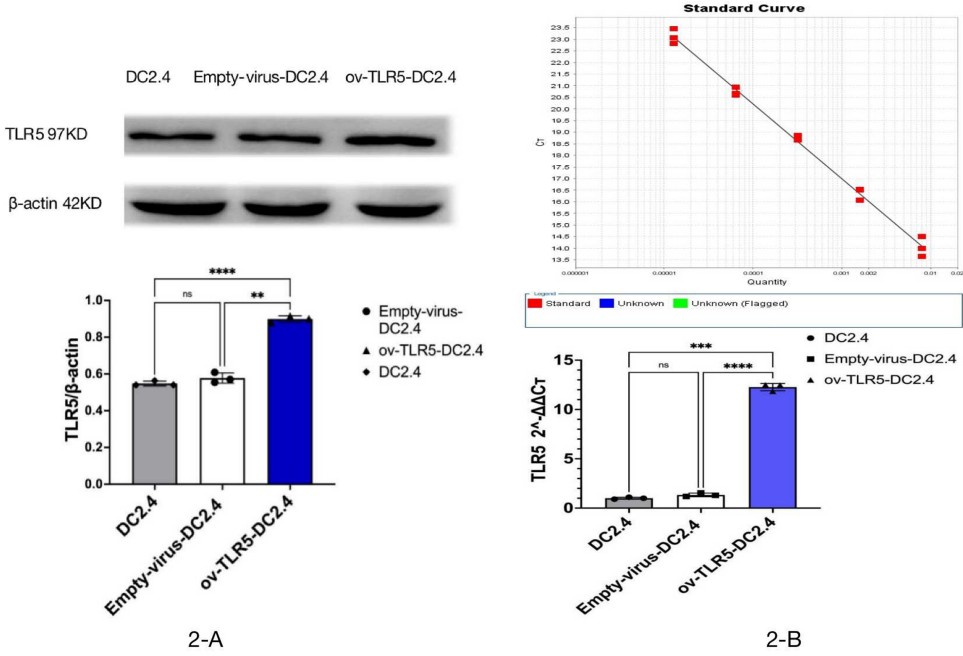

**Figure 2** **Construction of ov-TLR5-DC2.4 cell line.** (A) WB validation of ov-TLR5-DC2.4 TLR5 expression efficiency. (B) qPCR validation of ov-TLR5-DC2.4 TLR5 expression efficiency.

is inhibited, the induced TRIF protein and p-ERK1/2 protein by this complex stimulation signal will decrease.

## Proliferation of dendritic cells in each group

The proliferation activity of dendritic cells in the ov-TLR5-DC2.4+CBLB502 group was significantly enhanced compared to the other three groups, with statistical significance ($p = 0.0001$; $p < 0.0001$; $p = 0.0015$) as detected by CCK8 assay on the 5th day.

The DC2.4 cell line can enhance its cell proliferation ability under TLR5-flagellin complex signal stimulation, and the inhibition of the intracellular TRIF signal can suppress the cell proliferation ability induced by this complex signal, indicating that TRIF protein plays an important role in the cell proliferation process induced by TLR5-flagellin complex signal. See Fig. 4 for details.

## Changes in dendritic cell phenotype in each group

Using flow cytometry to detect the expression of surface markers MHCI, MHCII molecules, and co-stimulatory molecules CD80 and CD86 on dendritic cells in each group.These surface markers are considered to be the markers of maturation and enhanced antigen presentation ability of dendritic cell.

The results showed that compared with the other three groups, the expression of dendritic cell surface markers MHCI molecules, MHCII molecules, and co-stimulatory molecules CD80 and CD86 were up-regulated in the ov-TLR5-DC2.4+CBLB502 group

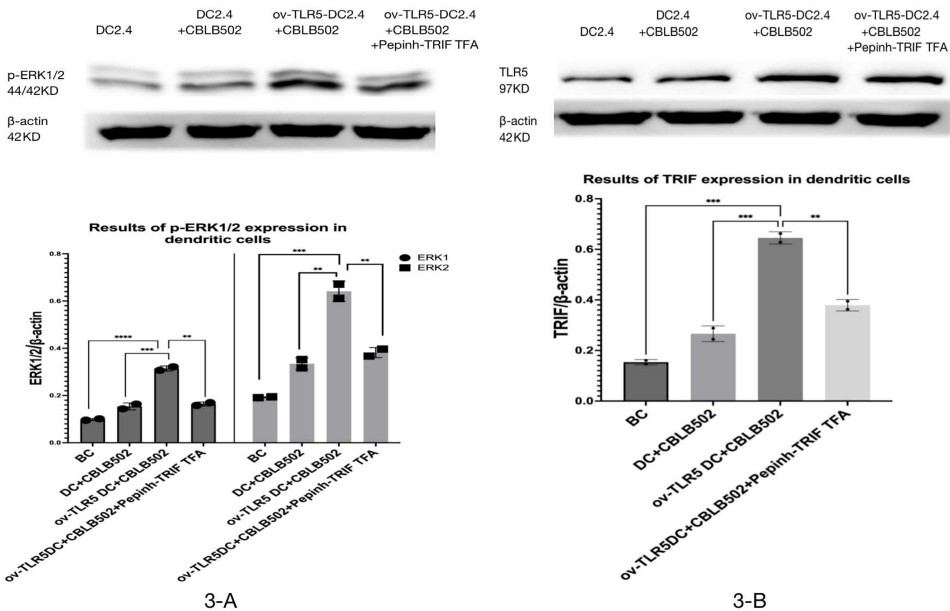

3-A            3-B

**Figure 3** **Expression of p-ERK1/2 protein in dendritic cells of each group and the effect of TRIF signal inhibition on its expression.** (A) Expression levels of TRIF protein in dendritic cells of each group. (B) Expression levels of p-ERK1/2 protein in each group treated.

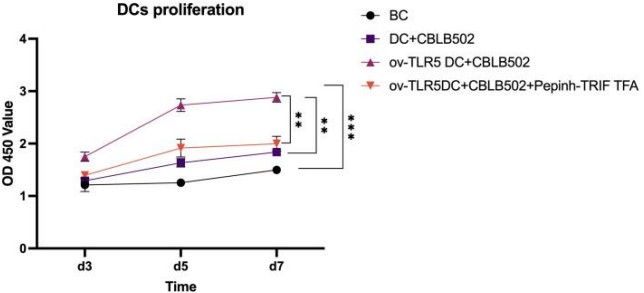

**Figure 4** **Proliferation of dendritic cells in each group.**

(MHCI: $p = 0.0010$, $p = 0.0027$, $p = 0.0062$; MHCII: $p = < 0.0001$, $p \leq 0.0001$, $p = 0.0003$; CD80: $p = 0.0019$, $p = 0.0004$, $p = 0.0005$; CD86: $p = 0.0025$, $p < 0.0001$, $p = 0.0128$).

The TLR5-flagellin protein complex signal can increase the maturity of dendritic cells and enhance their antigen presentation function. However, inhibiting the intracellular TRIF signal can suppress the antigen presentation function induced by this complex signal. See Fig. 5 for details.

## Secretion of IL-12 and IL-4 cytokines by dendritic cells in each group

IL-12 and IL-4 secretion in the cell culture supernatant of each group was detected using the ELSA method.

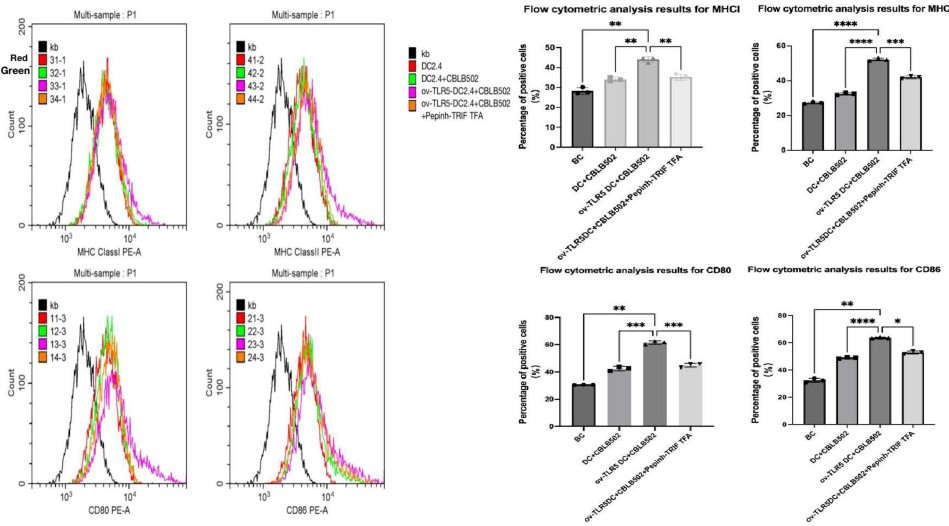

**Figure 5** **Expression of MHCI, MHCII, CD80, and CD86 molecules on the surface of dendritic cells in each group.**

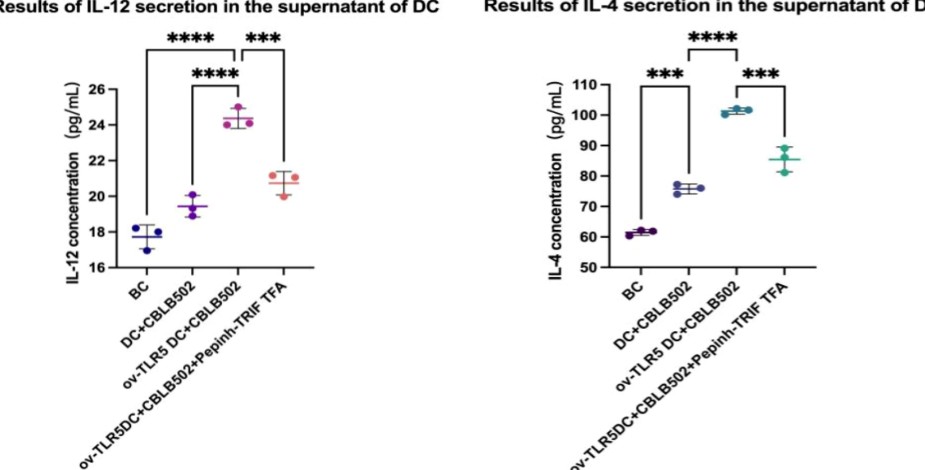

**Figure 6** **Secretion of IL-12 and IL-4 by dendritic cells in each group.**

The results showed that the contents of IL-12 and IL-4 in the ov-TLR5-DC2.4+CBLB502 group increased significantly compared with the other three groups (IL-12: $p < 0.0001$, $p < 0.0001$, $p = 0.0005$; IL-4: $p = < 0.0001$, $p = < 0.0001$, $p = 0.0001$).

The TLR5-flagellin complex signal group prompts dendritic cells to secrete proinflammatory cytokines such as IL-12 and IL-4. When the intracellular TRIF signal is inhibited, the secretion of inflammatory cytokines by dendritic cells can be suppressed, indicating that the TRIF protein plays an important role in dendritic cell-mediated mucosal inflammatory responses. See Fig. 6 for details.

## DISCUSSION

Through this study, we found that flagellin derivatives can selectively upregulate TRIF protein in myeloid dendritic cells with high expression of the TLR5 gene, and stimulate the upregulation of downstream proliferation-regulating protein p-ERK1/2, leading to enhanced proliferation activity of dendritic cells. The expression of surface MHCI molecules, MHCII molecules, and co-stimulatory molecules CD80 and 86 is upregulated, indicating enhanced antigen presentation ability of dendritic cells. At the same time, it further promotes the secretion of inflammatory cytokines IL-12 and IL-4 by dendritic cells. After inhibiting the TRIF signal in dendritic cells, the p-ERK1/2 signal is suppressed, accompanied by a decrease of surface markers MHCI molecules, MHCII molecules, and co-stimulatory molecules CD80 and CD86 expression in dendritic cell which represents the proliferation and antigen presentation abilities of dendritic cell were weakend. As well as a decrease in secretion of cytokines IL-12 and IL-4. Therefore, our results suggest that TRIF protein plays an important role in the proliferation and antigen presentation function of myeloid dendritic cells mediated by the flagellin protein-TLR5 complex signal.

In our previous animal experiments, we also found that an increase in intestinal mucosal flagellin protein content induced upregulation of intestinal mucosal TLR5 protein, TRIF protein, and phosphorylated p-ERK1/2 protein expression, and could activate LPDCs proliferation in the intestinal mucosa, accompanied by infiltration of inflammatory factors such as IL-4, IL-9, and IL-6 (*Zhuang et al., 2022*; *Li et al., 2015*; *Zhuang et al., 2016*).

In the immune response against pathogenic bacteria in the intestinal mucosa, previous studies mainly focused on the role of the Flagellin-TLR5 complex signal in the intracellular MyD88-dependent pathway. Currently, studies have shown that TRIF protein which is the intracellular adapter protein of TLRs also plays an important role in the regulation of the intestinal mucosal immune response initiated by TLR5. Studies have shown that TRIF protein can activate MAPK, NF-$\kappa$B protein, mediate the activation of signals such as JNK1/2 and ERK1/2, induce the production of a large number of chemokines and cytokines such as interleukin and tumor necrosis factor, and enhance the intercellular adhesion molecule-1 (ICAM-1) by upregulating the expression of inflammatory cells, leading to the abnormal mucosal immune response, imbalance of intestinal epithelial damage repair, and infiltration of inflammatory cells (*Zubair, Sad & Frieri, 2013*; *Bielaszewska et al., 2018*). Studies have shown that the silencing of the TRIF gene significantly reduces the activation of signals such as NF$\kappa$B (p105 and p65), JNK1/2, and ERK1/2 induced by flagellin protein, and reduces the expression of inflammatory cytokines in intestinal epithelial cells induced by flagellin protein (*Choi et al., 2010*).

ERK1/2 protein belongs to the mitogen-activated protein kinase cascade pathway signaling protein. Phosphorylated p-ERK1/2 can enter the nucleus and participate in the proliferation and differentiation of various cells, including dendritic cells, After maturation, dendritic cells can acquire extremely strong antigen presentation ability. They can provide T lymphocytes with the necessary antigen peptide MHCI/II signal for activation, as well as co stimulatory molecules such as CD80 and CD86 as second signals, inducing initial CD4+T lymphocytes to differentiate into Th1, Th2, Th17 and other cells, and initiating

corresponding immune responses, respectively, to produce some cytokines as well as the release of inflammatory factors, playing an important role in immune regulation (*Guindi et al., 2018*; *Jiang et al., 2018*; *Liu et al., 2016*; *Fucikova et al., 2019*).

Under steady-state conditions, lamina propria dendritic cells (LPDCs) in the intestinal mucosa exhibit an immature phenotype and have little ability to activate T lymphocytes, thus having low immunogenicity (*Lu et al., 2019*). However, after stimulation by flagellin-TLR5 complex signals. Under the coordinated action of the TRIF protein and the p-ERK1/2 signal, LPDCs surface markers transition towards mature phase, accompanied by an increase in antigen presentation ability. Subsequently promotes the secretion of cytokines IL-12 and IL-4. These pro-inflammatory factors secreted by dendritic cells have been shown to induce T lymphocyte differentiation and initiate mucosal immune response, subsequently promoting the release of inflammatory cytokines from T lymphocytes (*Bo et al., 2019*; *Yin, Chen & Eisenbarth, 2021*). Therefore, the maturation of LPDCs is crucial for the intestinal mucosal immune response and plays an important role in the subsequent infiltration of inflammatory factors in the intestinal mucosa (*Strauch et al., 2010*; *Mann et al., 2013*).

## CONCLUSION

Based on the above, we believe that flagellin protein upregulates intracellular TRIF protein, activates p-ERK1/2 signaling, promotes dendritic cell proliferation and antigen presentation function maturation by binding to the specific receptor TLR5 on myeloid dendritic cells, leading to pro-inflammatory cytokine secretion, mediating subsequent abnormal immune responses and inflammatory infiltration in the intestinal mucosa. Therefore, in diseases related to flagellin overgrowth caused by intestinal dysbiosis, the increase in flagellin protein content in the intestinal mucosa and the activation of flagellin protein-TLR5-TRIF-ERK1/2 signals require attention from researchers and can provide a theoretical basis for animal experiments and clinical research on such diseases.

We will further verify the specific mechanism of TRIF protein by animal experiments with this protein silenced.

### Funding
This work was funded by the Zhejiang Medical and Health Science and Technology Plan Project (2023KY1162); the Zhejiang Chinese Medical University School-level Scientific Research Project (2021FXYYZQ17); and the Project of Zhejiang Chinese Medical University (2021JKZKTS032B). The funders had no role in study design, data collection and analysis, decision to publish, or preparation of the manuscript.

### Grant Disclosures
The following grant information was disclosed by the authors:
Zhejiang Medical and Health Science and Technology Plan Project: 2023KY1162.

Zhejiang Chinese Medical University School-level Scientific Research Project: 2021FXYYZQ17.
Project of Zhejiang Chinese Medical University: 2021JKZKTS032B.

## Competing Interests

The authors declare there are no competing interests.

## Author Contributions

- Zhaomeng Zhuang conceived and designed the experiments, performed the experiments, analyzed the data, prepared figures and/or tables, and approved the final draft.
- Yi Chen analyzed the data, authored or reviewed drafts of the article, and approved the final draft.
- Juanhong Zheng performed the experiments, authored or reviewed drafts of the article, and approved the final draft.
- Shuo Chen conceived and designed the experiments, prepared figures and/or tables, and approved the final draft.

## Data Availability

The raw data is available in the Supplemental Files.

## Supplemental Information

Supplemental information for this article can be found online at http://dx.doi.org/10.7717/peerj.16716#supplemental-information.

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
