# Peer review of "The role of TRIF protein in regulating the proliferation and antigen presentation ability of myeloid dendritic cells through the ERK1/2 signaling pathway in chronic low-grade inflammation of intestinal mucosa mediated by flagellin-TLR5 complex signal"

_PeerJ, doi:10.7717/peerj.16716_

## Round 0.1 · original submission · Major Revisions

Please respond to the reviewer's comments and resubmit the revised one.

Reviewer 1 ·

Basic reporting

Several figures in the manuscript require significant adjustments:
- Figure 2:
- Enlarge the font size that describes the reaction conditions near WB.
- Increase the font sizes for the x, y axis labels, numbers, and the legends.
- Enhance the resolution of the standard curve plot.
- Figure 3:
- Font sizes describing the reaction conditions near WB are too small.
- Specify the number of duplicates/triplicates performed.
- For individual data points on the bar plot, reduce the symbols' size for better variance visualization.
- Figure 5:
- Figure 5A and 5B display the same information, but in different formats. Consider only presenting Figure 5B (overlay plot) to prevent redundancy.

Experimental design

Several errors are noted in this section:
- Line 85: What do you mean by “primer synthesis method”? Could it be "solid phase synthesis"?
- Line 92: Change "QT-PCR" to "qRT-PCR."
- Line 119: The micro-spectrophotometer measures RNA quantity, not quality.
- Lines 122-124: The microliter symbol's font is inconsistent with the rest. Kindly rectify.
- Line 124: '2' in 'ddH2O' should be subscripted.
- Figure 1: This seems more apt for supplementary information (SI) since it pertains to the methods. Please reconsider its placement.

Validity of the findings

- Figure 5: The results from the flow cytometry, as depicted in Figure 5, are not adequately discussed in the main text.
- General Observation: All figures are absent in the main text, leaving only their legends. Kindly rectify this oversight.

Reviewer 2 ·

Basic reporting

The authors have reported interesting phenomena where stimulating the Flagellin-TLR5 complex augments TRIF and p-ERK1/2 protein levels, leading to pro-inflammatory cytokines IL-12 and IL-4 secretion. However, more mechanistic insights into how these phenomena are correlated would increase the impact of the study.
The introduction needs to be more detailed. Most of the discussion can be used as an introduction for better understanding and increased readership.
In the Discussion, please discuss how your findings fill significant knowledge gaps.
In line 54: replace innate with inner.
In line 58-59 ‘In order to eliminate interfering factors, we designed this cell experiment’ please elaborate on the interfering factors.

Experimental design

Please present high-resolution images with readable legends.

Add space between number and symbol throughout the manuscript, e.g. 37_°C

Validity of the findings

For supplementary material, I could not open ‘Copy of results of pERK1:2 exp in DC’.


This paper cannot be considered for publication in its current form. However, it holds promising results, and I encourage the authors to revise and resubmit their manuscript.

Additional comments

Line 64: use the word antibiotics instead of antibodies.
Line 65: confluency

Line 70: the size of the amicon ultra mentioned needs to be corrected.
Line 122: Should be μM.

---

## Round 0.2 · Minor Revisions

Please response to all the reviewer's comments.

Reviewer 1 ·

Basic reporting

The standard curve is labeled with three categories: 'standard,' 'unknown,' and 'unknown (flagged);' however, the graph displays only red squares. Please ensure that the graph's visual elements align with the described categories in the legend.

Figure 5 presents two distinct sets of graphs. To improve clarity, assign and reference subnumbers to each graph and correlate these with the descriptions provided, enabling easier differentiation between the two groups.

Experimental design

To enhance the specificity of the procedure described in the article, it may be beneficial to refer to 'SOE PCR' instead of 'primer synthesis'.

Validity of the findings

The authors have not addressed the issues raised in this section.

Reviewer 2 ·

Basic reporting

All the asked changes have been made. The introduction has been changed and reads much better now. The overall flow of the paper has improved.

Experimental design

The resolution of figures has been improved, and legends are now readable. All the changes are made.

Validity of the findings

All the figure materials, including the supplementary material, are now readable.

Additional comments

The authors correctly addressed my feedback to improve the readability of the paper.

---

## Round 0.3 · accepted · Accept

Thanks for addressing all the comments from the reviewers.